# A Hybrid Method for Winter Road Surface Temperature Prediction Using Improved LSTMs and Stacking-Based Ensemble Learning

Wanting Li<sup>1,2,3</sup>, Linyi Zhou<sup>4,5</sup>, Xianghua Wu<sup>1,2,3</sup>, Miaomiao Ren<sup>1</sup>, Yuanhao Guo<sup>1</sup>, Kun Chen<sup>1</sup>, Huiwen Lin<sup>1</sup>

- 5 ¹ School of Mathematics and Statistics, Nanjing University of Information Science and Technology, Nanjing 210044, China ² Center for Applied Mathematics of Jiangsu Province, Nanjing University of Information Science and Technology, Nanjing 210044, China
  - <sup>3</sup> Jiangsu International Joint Laboratory on System Modeling and Data Analysis, Nanjing University of Information Science and Technology, Nanjing 210044, China
- O Annijing Innovation Institute for Atmospheric Sciences, Chinese Academy of Meteorological Sciences—Jiangsu Meteorological Service, Nanjing 210041, China
  - <sup>5</sup> Jiangsu Key Laboratory of SevereStorm Disaster Risk / Key Laboratory of Transportation Meteorology of CMA, Nanjing 210041, China
- 15 Correspondence to: Linyi Zhou (zhoulinyi@cma.gov.cn), Xianghua Wu (wuxianghua@nuist.edu.cn)

Abstract. Wintertime low temperatures and snow cover usually diminish the friction coefficient of asphalt pavements, thereby elevating accident and congestion risks. Road surface temperature (RST) is an important parameter for maintaining traffic safety under extreme winter weather conditions, as it helps predict road icing events. Aiming to enhance the precision and robustness of RST prediction, this paper introduces a forecasting framework combining optimized Long Short-Term Memory (LSTM) architectures with a stacking-based ensemble strategy. Two improved LSTMs are constructed: (1) KNN-LSTM, integrating Knearest neighbors to capture local spatiotemporal similarity patterns, and (2) Attention-BiLSTM, employing bidirectional temporal modeling with dynamic attention weighting mechanisms. These models function as base learners in the stacking ensemble, with Bayesian ridge regression utilized as the meta-learner to consolidate their predictions. The proposed hybrid model was trained and validated using minute-resolution winter meteorological data (2020-2024) collected from a station located on the Longhai Railway Bridge in Jiangsu, China. Experimental results show that the KNN-LSTM and Attention-BiLSTM models exhibit complementary advantages in capturing localized and global temporal features. The ensemble model demonstrates superior performance over individual models, achieving a 1-hour MAE of 0.074, MSE of 0.010, and MAPE of 46.7 % with a significant reduction compared with the best-performing single model. Under extended prediction horizons (3hour and 6-hour), including low-temperature below 0 °C conditions and typical weather backgrounds, the ensemble model sustains high prediction accuracy and stability. These findings underscore the efficacy of integrating local pattern extraction with attention-based mechanisms via ensemble learning, thereby enhancing RST prediction. This study presents a scalable and adaptable framework for intelligent road weather management systems, offering practical insights for operational deployment.

#### 1 Introduction





Road surface temperature is a critical parameter in both road safety and routine maintenance, as it directly affects pavement conditions and traffic efficiency (Horne, J. E., 1991; Shao and Lister, 1996). The intensification of climate change has led to more frequent extreme weather events, increasing the complexity of RST variation during winter due to amplified temperature fluctuations. The continuously changing winter conditions affect road maintenance operations and increase the risk of traffic accidents (Andersson and Chapman, 2011). Studies have shown a clear warming trend in northern China during winter, while certain southern regions have experienced colder conditions (Zhou et al., 2014), resulting in significant north-south disparities that pose new challenges for road management. In addition, shifts in precipitation patterns further complicate RST forecasting by altering surface thermal dynamics (Crevier and Delage, 2001).

The advent of Long Short-Term Memory networks marked a turning point, leveraging their capacity to model sequential data and long-term dependencies (Hochreiter and Schmidhuber, 1997). Subsequently, Gers et al. (2000) introduced the forget gate, enabling LSTM to "reset" its internal state and better retain long-sequence information. They further introduced peephole connections, allowing the gating mechanism to directly access cell state information, significantly improving model performance. In temperature-related prediction fields, LSTM has been applied to urban temperature forecasting and industrial temperature monitoring, effectively capturing long-term dependencies and improving prediction accuracy. For instance, Chen et al. (2023) proposed a real-time bridge temperature prediction method based on LSTM and ground meteorological data. Experimental results showed that the absolute prediction error of the LSTM model was within 2 °C, demonstrating good engineering applicability. Hou et al. (2021) investigated an LSTM recurrent neural network-based temperature prediction method for switchgear equipment, with prediction accuracy significantly higher than traditional BP neural networks and RNN models, indicating that LSTM has high stability and accuracy in industrial equipment temperature monitoring. Yan et al. (2024) proposed a temperature prediction model combining convolutional neural networks and long short-term memory LSTM for real-time global temperature prediction, validating that the CNN-LSTM model has higher accuracy and generalization capability when processing complex meteorological data.

The field of RST forecasting has also witnessed substantial advancements, evolving from traditional statistical models to sophisticated deep learning techniques. Early approaches, such as linear regression and basic time-series analyses, struggled to capture the nonlinear and temporally dependent nature of RST data (Chapman et al., 2001). The emergence of machine learning and deep learning methods has provided new solutions to overcome these limitations. Nowrin and Kwon (2022) applied artificial neural networks to predict RST using data from Road Weather Information System (RWIS) stations, highlighting the sensitivity of prediction accuracy to forecast horizons and geographical features. In the same period, regional implementations showed promising results, as demonstrated by Xie et al. (2023) in Hunan Province, where logistic regression paired with thermal mapping enabled an RST-based icing warning system with enhanced spatial resolution. Dai et al. (2023) optimized a multidimensional LSTM model by integrating meteorological cumulative effects and temperature periodicity through a sliding window approach. Compared to Random Forest (RF) and Backpropagation (BP) neural networks, the LSTM-based RST hourly

75

prediction model demonstrated significant advantages in both accuracy and stability, proving the strong generalization capability and robustness of LSTM in complex heat transfer processes. Most recently, hybrid models have shown exceptional performance in complex geographical conditions. Zhang et al. (2024) developed a hybrid RF-LSTM model specifically designed to address the challenging climatic conditions in high-altitude mountainous areas. By integrating feature selection via the random forest algorithm with time-series modeling using LSTM, the proposed model achieves a prediction accuracy of 99.13 %, maintaining prediction errors within  $\pm$  0.5 °C. These results demonstrate strong robustness and transferability of the model, particularly in mountainous regions. The research outcomes presented above indicate that deep learning techniques, especially improved models based on LSTM, have achieved notable technological advancements and possess substantial potential for practical application in predicting road surface temperatures.

Ensemble learning techniques, notably stacking, a layered method introduced by Wolpert (1992), have emerged as powerful tools for improving model generalization by achieving model fusion through hierarchical training of base learners and metalearners In subsequent research, Ledezma et al. (2010) proposed a genetic algorithm-based stacking optimization framework (GA-Stacking), which significantly improved classification task accuracy through dynamic adjustment of base model weights. The advantages of stacking became further pronounced in time series prediction, as demonstrated by Assaad et al. (2008), who combined Boosting with recurrent neural networks and improved long-term prediction stability through stacking ensemble strategies. Building upon these foundations, Zheng et al. (2023) integrated the GBDT algorithm with stacking strategies, enhancing the accuracy and stability of short-term power load forecasting through multi-level ensemble optimization.

In recent years, the combination of deep learning methods with ensemble strategies has emerged as a prominent research trend. LSTM, owing to its excellent temporal modeling capabilities, is frequently incorporated as a base learner in ensemble frameworks. Yang et al. (2010) proposed a weighted clustering ensemble-based approach for time series data modeling, reducing prediction errors by fusing multiple LSTM sub-models. Subsequently, Yang et al. (2022) developed a short-term wind power prediction model based on Attention-GRU wind speed correction and stacking multi-algorithm fusion, significantly improving both the accuracy and generalization capability of wind power forecasting. Furthermore, the introduction of attention mechanisms has further optimized ensemble effects, as exemplified by Guo et al. (2013), who combined attention mechanisms with stacking in medical image analysis to enhance feature selection precision.

Meteorological data is characterized by strong temporal dependencies and substantial noise interference, causing traditional single models to frequently encounter overfitting or underfitting problems. Ensemble learning has emerged as an effective solution to address such challenges by integrating the advantages of multiple models. Subramanian et al. (2011) employed ensemble models to analyze multi-source time series data, successfully predicting mortality rates in heart failure patients and validating the potential of ensemble methods in complex temporal modeling. Similarly, Li et al. (2024) established a meteorological factor-based corn yield prediction model by analyzing the importance of meteorological factors and utilizing three ensemble learning methods: LightGBM, bagging, and stacking. The results demonstrated that the stacking model exhibited optimal performance in terms of both prediction accuracy and robustness, effectively leveraging the characteristics and advantages of individual base learners.

This study proposes a winter road surface temperature prediction model based on improved LSTMs and stacking integration. The approach enhances feature selection through the incorporation of the K-nearest neighbors algorithm, strengthens temporal feature extraction using an attention mechanism, and designs an efficient meta-learner fusion strategy to improve prediction accuracy and model robustness. The model is trained and validated using RST and meteorological data from a road weather station in Jiangsu Province. After data preprocessing and variable correlation analysis, we design the model architecture to capture RST periodicity and the lag effects of meteorological inputs. Our approach provides a scalable and adaptive framework for RST forecasting and supports the development of early-warning systems in cold-region road management.

## 2 Methodology




Fig. 1 illustrates the working mechanism of the improved LSTM-stacking ensemble model for winter road surface temperature prediction, structured into three distinct stages: data processing and feature extraction, model training, and predictive performance evaluation. In the first stage, historical meteorological data relevant to road surface temperature are acquired. The data undergo a series of quality control steps, including cleaning, outlier removal, and missing value imputation. Once the data is processed, five variables that most accurately reflect the characteristics of winter road surface temperature variations are selected as model inputs based on the Spearman rank correlation coefficient. A feature matrix is constructed, with each feature representing the relevant meteorological parameters and their historical corresponding values. This stage ensures the preparation of high-quality data for the subsequent modeling phase. The second stage focuses on the training of the constructed improved LSTM-stacking ensemble model. Improved LSTMs serve as base learners in this ensemble approach: the KNN-LSTM integrates the K-nearest neighbors algorithm with the sequential modeling capability of LSTM to capture local similarity patterns, whereas the Attention-BiLSTM incorporates a bidirectional temporal modeling structure along with an attention mechanism to effectively identify critical temporal features. The Bayesian ridge regression model functions as a meta-learner, integrating the outputs from the base models and performing further optimization to generate the final RST prediction. In the final stage, a comprehensive evaluation of the improved LSTM-stacking ensemble model is performed by comparing it with LSTM, KNN-LSTM, and Attention-BiLSTM. Model performance is assessed using multiple metrics, including prediction accuracy, stability, and adaptability under varying prediction horizons, extreme temperature scenarios, and typical weather conditions. Each of the above steps is described in detail below.

Figure 1: Overview of the improved LSTM-stacking ensemble model for winter road surface temperature prediction.

## 2.1 KNN-LSTM



The KNN-LSTM model combines the strengths of KNN (Cover and Hart, 1967) and LSTM networks to enhance predictive accuracy by integrating similarity-based feature augmentation with temporal sequence modeling. Originally proposed by Luo et al. (2019) for traffic flow prediction, this approach is extended here for wintertime road surface temperature forecasting. The modeling process is outlined below:

For a given target time step t, the input meteorological feature vector is denoted as:  $X_t = (X_{t,1}, X_{t,2}, ..., X_{t,n})$ , where n is the feature dimension, and each component  $X_{t,j}$  corresponds to the value of the j-th feature at time t. Let  $X_i = (X_{i,1}, X_{i,2}, ..., X_{i,n})$  be the i-th historical sample in the training set, where i = 1, 2, ..., M and M is the total number of training samples. The similarity between the input vector  $X_t$  and a historical vector  $X_i$  is calculated using the Euclidean distance:

$$d(X_t, X_i) = \sqrt{\sum_{j=1}^n (X_{tj} - X_{ij})^2},$$
(1)

The K nearest neighbors with the smallest Euclidean distances are selected to form a similarity set  $\{X_i\}_{i=1}^K$ , where each  $X_i$  denotes the input feature vector of the i-th nearest historical sample. A similarity feature vector  $F_t$  is then constructed by inverse-distance weighted averaging of the K neighbors:

$$F_t = \frac{\sum_{i=1}^K \frac{1}{d(X_t, X_i)} X_i}{\sum_{i=1}^K \frac{1}{d(X_t, X_i)_i}},$$
 (2)


where  $d(X_t, X_i)_i$  is the Euclidean distance between the current input  $X_t$  and the *i*-th neighbor  $X_i$ . The resulting vector  $F_t$  represents the mean of the most similar historical patterns, weighted by proximity.

The original input vector  $X_t$  and the similarity-enhanced vector  $F_t$  are then concatenated along the feature dimension to form an augmented 2n-dimensional feature vector:  $X_{t,aug} = (X_t, F_t) = (X_{t,1}, X_{t,2}, \cdots, X_{t,n}, F_{t,1}, F_{t,2}, \cdots, F_{t,n})$ . To capture temporal dependencies, a sliding window of size T is applied to the augmented input sequence, producing a 3D tensor for LSTM input:  $X_{lstm} \in R^{M \times T \times 2n}$ .

The augmented sequence  $X_{lstm}$  is then passed into an LSTM network, which models temporal dependencies through its gated architecture. After processing T time steps, the hidden state at the final time step  $h_{t,T}$ , is extracted. This hidden state captures the long-term dependencies embedded in the input sequence and serves as a compact representation of the temporal dynamics of the system.

The final hidden state  $h_{t,T}$  is passed through a fully connected output layer to generate the predicted road surface temperature:

$$\hat{y}_{knn} = W_0 \cdot h_{t,T} + b_0 , \qquad (3)$$

where  $W_0$  is the weight vector and  $b_0$  is the bias term. The output  $\hat{y}_{knn}$  is the predicted road surface temperature at time t+1.

After each prediction, the value of K is incremented by 1, and the next most similar sample is considered for augmentation.

The loop continues until the value of K reaches M, the total number of training samples. A schematic overview of the complete KNN-LSTM architecture is provided in Fig. 2.

Figure 2: Architecture of the KNN-LSTM model.

## 2.2 Attention-BiLSTM

The Attention-BiLSTM model integrates Bidirectional Long Short-Term Memory networks with a self-attention mechanism (Bahdanau et al., 2014), enabling the network to simultaneously capture temporal dependencies in both forward and backward directions and assign adaptive importance to different time steps. As proposed by Zhou Wenye et al. (2019), the Attention-BiLSTM architecture enhances the traditional LSTM by assigning varying weights to different time steps, allowing the model to focus more effectively on key information within the input sequence. The model structure, illustrated in Fig. 3, proceeds as follows:

Let the sliding window size be T, and the input sequence at time t is specifically represented as:  $X = (X_{t-T+1}, X_{t-T+2}, \dots, X_t)^T$ ,  $X_t = (X_{t,1}, X_{t,2}, \dots, X_{t,n})$ , where n is the feature dimension. First, a bidirectional LSTM is used to extract temporal features from the input sequence (Schuster and Paliwal, 1997). The forward LSTM processes the sequence from past to future, while the backward LSTM captures future-to-past dependencies. At time step t, the output state  $\overrightarrow{h_{t,l}}$  of the t-th input vector in the forward layer is computed using the input sequence t, while the output state t-th backward layer is computed using the reversed form of t-th input vector in the forward layer at time t-th input vector in the forward layer is computed as t-th input vector in the forward layer is computed using the reversed form of t-th input vector in the forward layer at time t-th input vector in the forward layer is computed using the reversed form of t-th input vector in the forward layer at time t-th input vector in the forward layer is computed using the reversed form of t-th input vector in the forward layer is computed using the reversed form of t-th input vector in the forward layer at time t-th input vector in the forward layer is computed using the reversed form of t-th input vector in the forward layer is computed using the reversed form of t-th input vector in the forward layer is computed using the reversed form of t-th input vector in the forward layer is computed using the reversed form of t-th input vector in the forward layer is computed using the reversed form of t-th input vector in the forward layer is computed using the reversed form of t-th input vector in the forward layer is computed using the reverse formation t-th input vector in the forward layer is computed using the reverse formation t-th input vector in the forward layer is computed using the reverse formation t-th input vector in the forward layer is computed using the reve

For each time step, the attention mechanism calculates an attention score  $\alpha_{t,i}$ , reflecting the importance of each hidden state  $h_t$  for the final prediction. The attention score is computed using a dot-product similarity function:

$$\alpha_{t,i} = \frac{exp(score(h_t, h_i))}{\sum_{k=1}^{T} exp(score(h_t, h_k))},$$
 (4)

where the *score* function measures the similarity between the current time step t and each of the previous time steps using the dot product between the hidden states.

$$score(h_t, h_i) = h_t \cdot h_i$$
, (5)

The weighted context vector  $c_t$  is then obtained by summing the weighted hidden states:

$$c_t = \sum_{i=1}^{T} \alpha_{t,i} h_{t,i}$$
, (6)

This context vector  $c_t$  is subsequently used in predicting the future RST. The hidden state  $h_t$  and the context vector  $c_t$  are combined to generate the final output. This combined vector is passed through a fully connected layer to yield the final temperature prediction:

$$\hat{y}_{att} = W_0[h_t, c_t] + b_0 \,, \tag{7}$$

where  $W_0$  is the weight matrix of the fully connected layer, and  $b_0$  is the bias term.

Figure 3: Architecture of the Attention-BiLSTM model. Here,  $\overrightarrow{h_{t,i}}$  is the forward hidden state at time step t for the i-th input vector,  $h_{t,i}$  is the backward hidden state at time step t for the i-th input vector,  $h_{t,i}$  is the combined output of the forward and backward hidden states,  $\alpha_{t,i}$  is the attention weight at time step t for the i-th input vector.

## 190 2.3 Stacking ensemble learning strategy


Stacking is a hierarchical ensemble learning method designed to enhance prediction performance through the integration of outputs from multiple individual models. In contrast to bagging and boosting, which rely on parallel or sequential resampling strategies, stacking adopts a multi-layer architecture consisting of base learners in the first layer and a meta-learner in the second layer. As illustrated in Fig. 4, this layered structure enables stacking to effectively handle complex learning tasks, making it particularly suitable for applications requiring the combination of diverse model capabilities.

Figure 4: Hierarchical structure of stacking.



In the stacking framework, base learners and meta-learner have clear roles and collaborate to make predictions. Base learners, located in the first layer, directly process the raw input and generate preliminary prediction results. To enhance model diversity, models with structurally different characteristics are typically chosen, which extract feature information from various perspectives. Such diversity is crucial for enhancing the effectiveness and generalization ability of the stacking ensemble.

The meta-learner, located in the second layer, is responsible for integrating the outputs of the base models and learning the mapping relationship between them and the true values. During training, the meta-model uses the results of cross-validation from the base learners as input for supervised learning, thus capturing the complementarity between the models. Through this hierarchical collaboration mechanism, stacking effectively combines the advantages of multiple models, enhancing the accuracy and stability of predictions.

## 2.4 Construction of the improved LSTM-stacking ensemble model

The selection of appropriate base learners and a suitable meta-learner is essential for enhancing the predictive performance of stacking-based ensemble models. In this study, KNN-LSTM and Attention-BiLSTM are selected as base learners. Both models are variants of the LSTM network and are capable of effectively modeling nonlinear characteristics and capturing long-term dependencies inherent in time series data. The KNN-LSTM model integrates the local search capability of the K-nearest



neighbor algorithm with the temporal modeling capacity of the LSTM network to identify local patterns. Meanwhile, the Attention-BiLSTM model incorporates an attention mechanism, which dynamically assigns temporal weights, thereby facilitating the extraction of critical temporal features. Together, these two base learners provide comprehensive prediction information from complementary perspectives, forming a robust foundation for the ensemble approach.

Given that the predictions from KNN-LSTM and Attention-BiLSTM already encapsulate complex data patterns, the role of the meta-learner becomes to model the combination of these predictions in a way that minimizes the final prediction error. In this study, we select Bayesian ridge regression as the meta-learner due to its ability to handle multicollinearity between the predictions from the base models and its regularization properties, which help prevent overfitting. Bayesian ridge regression works by performing a weighted sum of the predictions from both base learners, with the weights determined through optimization. This allows the model to assign different contributions to each base learner based on the specific data scenario, reflecting their relative strengths and weaknesses. The Bayesian nature of the model also provides a probabilistic framework, enabling it to model uncertainty in the predictions from the base learners and incorporate this uncertainty into the final ensemble prediction. This flexibility ensures that the final model not only produces accurate predictions but also accounts for the inherent variability present in the outputs of the base models. The structural overview of the proposed improved LSTM-stacking ensemble model is presented in Fig. 5.

Figure 5: Architecture of the improved LSTM-stacking ensemble model.



# 3 Experimental details

## 3.1 Data and preprocessing

This study focuses on a segment of the Longhai Railway Bridge situated within the northwest inland plain of Jiangsu Province, China. The region falls within the typical warm temperate semi-humid monsoon climate zone. The topography is primarily flat and expansive, which facilitates the horizontal distribution and transmission of meteorological factors. The data used in this study were collected from a minute-level observational meteorological station located to the west of the Longhai Railway Bridge in Jiangsu Province, at coordinates (34.30°N, 117.04°E). The dataset spans four periods: December 2020 to February 2021, December 2021 to February 2022, December 2022 to February 2023, and December 2023 to February 2024. During this period, infrared remote sensing sensors collected data every 5 minutes, recording ground-level meteorological variables including visibility, air temperature, relative humidity, rainfall, wind speed, wind direction, and RST, with a total of over 100,000 data points collected. Due to missing data, solar radiation was not considered in this study. After data cleaning, imputation, and outlier handling, the meteorological factors and RST data were resampled at hourly intervals, resulting in a refined dataset of 8,544 samples for modeling. A detailed summary of the dataset is provided in Table 1.

Table 1: Summary of the dataset.

| Variable          | Description                                                                                        | Mean    | Standard deviation | Min    | Max      |
|-------------------|----------------------------------------------------------------------------------------------------|---------|--------------------|--------|----------|
| Visibility        | Maximum distance (m) that a person with normal vision can recognize the target.                    | 7942.28 | 3355.89            | 56.00  | 30000.00 |
| Air temperature   | Temperature (°C) indicated by a thermometer exposed to the air but sheltered from direct sunlight. | 2.08    | 5.29               | -15.15 | 23.79    |
| Relative humidity | Ratio (%) of actual water vapor pressure to saturation water vapor pressure.                       | 55.97   | 24.14              | 1.76   | 100.00   |
| Rainfall          | Depth (mm) to which rainwater accumulates on the surface in 1-hour.                                | 0.01    | 0.12               | 0.00   | 4.60     |
| Wind speed        | Rate (m·s <sup>-1</sup> ) at which air is moving.                                                  | 1.89    | 1.19               | 0.30   | 8.72     |
| Wind direction    | Direction (°) from which the wind is coming.                                                       | 175.31  | 78.63              | 4.50   | 350.00   |
| RST               | Road surface temperature (°C) indicated by the infrared remote sensor.                             | 3.72    | 5.59               | -12.99 | 26.52    |

#### 3.2 Feature extraction

Due to the material properties of asphalt pavement, its temperature is primarily driven by meteorological factors, exhibiting significant periodic lag variations. This is a complex, nonlinear heat transfer process (Chen et al., 2019). Fig. 6 illustrates the daily periodic variation of RST. The fluctuation in RST ranges from approximately 5 °C to 10 °C daily, with a strong correlation observed at the same time each day. Considering that RST is influenced both by the periodic rhythm driven by the day-night cycle and by random fluctuations caused by meteorological disturbances, this study incorporates historical RST data into the prediction model to simulate its periodicity.

Figure 6: Periodic variation of road surface temperature. T represents a period, with time corresponding to one day.

To reduce model complexity and improve prediction accuracy, Spearman rank correlation coefficients (Spearman, 1961) were used to quantify the monotonic relationships between meteorological variables and winter RST. This nonparametric metric ranges from -1 to 1 and is suitable for assessing non-linear dependencies. It is calculated as:

$$\rho = 1 - \frac{6\Sigma d_i^2}{n(n^2 - 1)},$$
 (8)

where  $d_i$  is the rank difference between paired observations, and n is the number of samples.

As illustrated in Fig. 7, air temperature had the strongest positive correlation with RST, with a correlation coefficient of 0.93. Wind speed and precipitation also displayed moderate positive correlations of 0.29 and 0.27, respectively. In contrast, relative humidity was negatively correlated with RST, with a coefficient of -0.25, likely due to its dampening effect on evaporative cooling. Visibility and wind direction showed weak correlations, both below 0.2. Based on these findings, the five selected

predictors for the model were air temperature, relative humidity, precipitation, wind speed, and lagged RST.

Figure 7: Spearman correlation coefficient plot between RST and various meteorological factors.

# 3.3 Optimization


## 265 3.3.1 Sliding window size

The sliding window method is employed to construct input variables and prediction targets for the model, and the selection of the window size directly influences the ability of the model to capture temperature periodicity and lag effects of meteorological variables. By comparing the performance of the models under various window sizes, the results indicate that when the sliding window size is set to 24 hours, the KNN-LSTM and Attention-BiLSTM models both achieve optimal performance based on all evaluation metrics. Specifically, at the 24-hour window, the MAE and MAPE of the KNN-LSTM model are 0.093 and 61.3 %, respectively, while the Attention-BiLSTM model yields an MAE of 0.093 and a MAPE of 54.7 %. Both models attain a MSE

of 0.015. Consequently, this study selects a 24-hour sliding window size as the best choice to balance accuracy and computational cost.

Table 2: Impact of sliding window size on errors

| Window size | Model            | MAE                        | MSE                                                                    | MAPE (%) |
|-------------|------------------|----------------------------|------------------------------------------------------------------------|----------|
| <i>(</i> 1, | KNN-LSTM         | 0.106                      | 0.019                                                                  | 85.2     |
| 6-hour      | Attention-BiLSTM | 0.114                      | 0.019<br>4 0.022<br>0.015<br>0.015<br>0.015<br>0.016<br>0.016<br>0.016 | 51.0     |
| 24-hour     | KNN-LSTM         | 0.093 0.013<br>0.093 0.013 | 0.015                                                                  | 61.3     |
| 24-110ur    | Attention-BiLSTM | 0.093                      | 0.019<br>0.022<br>0.015<br>0.015<br>0.016<br>0.016                     | 54.7     |
| 48-hour     | KNN-LSTM         | 0.094                      | 0.016                                                                  | 68.8     |
| 46-flour    | Attention-BiLSTM | 0.097                      | 0.022<br>0.015<br>0.015<br>0.016<br>0.016<br>0.017                     | 74.1     |
| 72-hour     | KNN-LSTM         | 0.098                      | 0.019<br>0.022<br>0.015<br>0.015<br>0.016<br>0.016                     | 69.1     |
| / 2-110uf   | Attention-BiLSTM | 0.114                      |                                                                        | 71.2     |

## 275 **3.3.2** *K* value


In the KNN-LSTM model, the parameter K determines the number of historical samples utilized for constructing similarity-based features, thereby directly influencing the capability of the model to capture local patterns. A smaller value of K may lead to overfitting, as the model becomes excessively dependent on local information, whereas a larger value of K could obscure important details and diminish the sensitivity of the predictions. To identify the optimal value of K, cross-validation is conducted to evaluate the MAE of the model on the validation dataset across different values of K. As shown in Fig. 8, the MAE decreases progressively with an increasing K, reaching its lowest value at K = 15, after which it stabilizes. This suggests that setting K to 15 achieves an effective balance between prediction accuracy and stability, ensuring robustness while mitigating the risk of overfitting.

#### Figure 8: The effect of K value on MAE

#### 3.3.3 Training


The dataset was divided into training and testing subsets. The training subset comprised data collected from December 1, 2020, to February 28, 2023, whereas the testing subset included data collected from December 1, 2023, to February 29, 2024. Prior to being input into the model, all variables were standardized using the Z-score normalization method to accelerate model convergence and mitigate the influence of differing measurement units among input variables on the computational complexity of the model:

$$X_{scaled} = \frac{X - \mu}{\sigma},\tag{9}$$

where  $X_{scaled}$  is the newly standardized data.

temperature prediction, several important parameters are carefully tuned in this study. Within the LSTM architecture, the first and second layers of the KNN-LSTM model contain 64 and 32 units, respectively, whereas the corresponding layers of the Attention-BiLSTM model consist of 128 and 64 units, respectively, in order to attain superior predictive accuracy. The Adam optimizer is selected with a learning rate of 0.001, and early stopping is implemented to terminate training when the validation loss shows no improvement for 20 consecutive epochs. To mitigate the vanishing gradient issue, the tanh and ReLU activation functions are utilized within different layers of the model. During the training phase, the batch size is set at 32 with a maximum epoch number of 50. Additionally, to reduce the risk of overfitting, a dropout rate of 0.2 and L2 regularization are applied following the LSTM layers. The MSE loss function is adopted to optimize the parameters of the model. These strategies effectively enhance the generalization capability of the model and improve predictive performance when applied to unseen data.

# 3.4 Evaluation

In this study, multiple evaluation metrics are employed to comprehensively evaluate the predictive performance of the models, including the mean absolute error (MAE), mean squared error (MSE), mean absolute percentage error (MAPE), and coefficient of determination (R<sup>2</sup>).

$$MAE = \frac{1}{n} \sum_{i=1}^{n} |y_i - \hat{y}_i|,$$
 (10)

$$MSE = \frac{1}{n} \sum_{i=1}^{n} (y_i - \hat{y}_i)^2,$$
(11)

MAPE = 
$$\frac{1}{n} \sum_{i=1}^{n} \left| \frac{y_i - \hat{y}_i}{y_i} \right| \times 100$$
, (12)

$$R^{2} = 1 - \frac{\sum_{i=1}^{n} (y_{i} - \hat{y}_{i})^{2}}{\sum_{i=1}^{n} (y_{i} - \bar{y})^{2}},$$
(13)

where  $y_i$  is the observed value,  $\hat{y}_i$  is the predicted value,  $\bar{y}$  is the mean of the observed values, and n is the total number of samples.

#### 4 Results







## 4.1 Winter RST prediction at different time intervals

## 4.1.1 Model fitting performance

An improved LSTM-stacking ensemble model incorporating KNN-LSTM and Attention-LSTM as base learners with a Bayesian ridge regression meta-learner was developed for winter RST prediction. Model performance in RST forecasting demonstrates significant variability in response to different prediction intervals. This study systematically investigates the generalization capabilities and relative merits of various algorithms at multiple temporal resolutions (1-, 3-, and 6-hour intervals), with special focus on the long-term predictive performance advantages of the ensemble model The density scatter plots in Fig. 9 are analyzed through both cross-temporal horizontal and cross-model vertical perspectives to evaluate predictive accuracy, goodness-of-fit metrics, and density distribution patterns.

Horizontally, across different time intervals, all models demonstrate superior performance for shorter prediction intervals. In the 1-hour predictions, observed and predicted values exhibit close alignment, with scatter points concentrated near the ideal y = x line and yellow-indicated high-density regions reflecting a strong model fit. As the prediction interval extends to 3 and 6 hours, the scatter points exhibit progressive dispersion, characterized by larger errors, more outliers, and reduced high-density areas, while blue-indicated low-density regions become dominant, leading to a noticeable decline in predictive performance. This suggests that models are more effective at capturing short-term temperature variations, while accuracy and stability decrease in longer-term predictions, as further corroborated by the sparsening of density distribution.

Vertically, across different models, the LSTM-stacking ensemble model demonstrates consistent superiority across all temporal intervals. For 1-hour, 3-hour, and 6-hour predictions, its scatter points exhibit the most compact clustering, with high-density regions concentrated near the ideal y = x line, achieving an  $R^2$  value of 0.992 for the 1-hour interval. Even in 6-hour predictions, the ensemble model effectively preserves temperature trends, maintaining an  $R^2$  value of 0.766 with a relatively concentrated density distribution. Comparatively, the conventional LSTM model demonstrates the weakest fitting performance, particularly for the 6-hour interval, where its scatter points show the broadest dispersion, low-density regions predominate, and the  $R^2$  value drops to 0.638, indicating substantially reduced predictive accuracy.

Furthermore, density distribution analysis reveals that scatter points exhibit progressive dispersion at elevated temperatures, characterized by a gradual transition from high-density to low-density regions as temperature increases. This indicates a systematic increase in prediction errors and enhanced error variability under high-temperature conditions. This trend becomes particularly evident in 6-hour predictions, where the number of outliers in the high-temperature range rises markedly,

accompanied by a pronounced reduction in density across all four models. This phenomenon may be attributed to the more complex variations in RST under high-temperature conditions.

Figure 9: Density scatter plots of predicted versus observed winter RST across 1-hour (a, d, g, j), 3-hour (b, e, h, k), and 6-hour (c, f, i, l) forecasting intervals. Here, colors indicate the density distribution, with yellow representing high-density regions and blue representing low-density regions. The term "Ensemble" in panels (i), (k), and (l), as well as subsequent figures, refers to the proposed improved LSTM-stacking ensemble model.

## 4.1.2 Sensitivity analysis





Taking January 2024 as a case study, Fig. 10 illustrates time-series comparisons of winter RST predictions generated by different models across multiple prediction intervals. For 1-hour forecasts, all algorithms demonstrate effective trend capture capabilities. Notably, during phases characterized by pronounced or periodic thermal oscillations, exemplified by the period spanning January 7-13, models exhibit rapid responsiveness to transient temperature variations, with predicted values maintaining close alignment with observed RST data.

Compared to the 1-hour forecasts, the 3-hour prediction trajectories exhibit smoother characteristics while sustaining satisfactory predictive performance. However, during phases marked by relatively stable or stochastic temperature fluctuations, exemplified by midday and nighttime periods spanning January 16-20, the prediction deviations in individual models tend to amplify. Conversely, the prediction trajectory of the LSTM-stacking ensemble model demonstrates closer alignment with observed winter RST values, exhibiting enhanced adaptive capacity under such conditions.

When the prediction interval is extended to 6 hours, the attenuated amplitude of the prediction trajectories indicates a compromised capacity of models to track transient thermal variations. Within this interval, the conventional LSTM model demonstrates significantly amplified prediction errors, particularly during phases of pronounced nocturnal thermal oscillations, where deviations reach their maximum magnitude. Conversely, the LSTM-stacking ensemble model maintains robust stability across all temporal scales, effectively suppressing the error fluctuations inherent in individual models while delivering superior predictive accuracy.

Figure 10: Time-series plots of winter RST predictions across 1-hour (a), 3-hour (b), 6-hour (c) forecasting intervals.

## 4.1.3. Error distribution analysis


With respect to quantitative error metrics, the LSTM-stacking ensemble model consistently exhibits superior performance. Across all temporal scales and evaluated error metrics, the ensemble framework surpasses all individual models. For 1-hour forecasts, the model achieves a MAE of 0.074, a MSE of 0.010, and a MAPE of 46.7 %, corresponding to reductions of 20.4 %, 33.3 %, and 14.6 % relative to the best-performing single model. At the 3-hour prediction interval, the error magnitude of the




ensemble model escalates, recording aMAE of 0.229, a MSE of 0.102, and a MAPE of 125.9 %, while maintaining superior performance over individual models. The performance of KNN-LSTM and Attention-BiLSTM models at the 3-hour prediction interval demonstrates comparable efficacy, with both variants outperforming the conventional LSTM model. Moving to the 6-hour forecasting interval, all models exhibit a pronounced degradation in predictive performance. At this extended interval, the improved LSTM-stacking ensemble model achieves a MAE of 0.380, marking a 19.8 % improvement over the LSTM model. The model registers a MSE of 0.270, equivalent to 77.8 % of the value observed for the Attention-BiLSTM variant. The MAPE reaches 187.3 %, reflecting a 24.8 % reduction relative to the second-best model. In contrast, the MAPE of the LSTM model escalates to 310.2 %, representing a 344.9 % increase compared to its 1-hour forecast value.

Table 3: Performance evaluation across 1-hour, 3-hour, 6-hour forecasting intervals.

| Time   | Index    | LSTM  | KNN-LSTM | Attention-BiLSTM | Ensemble |
|--------|----------|-------|----------|------------------|----------|
|        | MAE      | 0.100 | 0.093    | 0.093            | 0.074    |
| 1-hour | MSE      | 0.021 | 0.015    | 0.015            | 0.010    |
|        | MAPE (%) | 69.7  | 61.3     | 54.7             | 46.7     |
|        | MAE      | 0.313 | 0.258    | 0.278            | 0.229    |
| 3-hour | MSE      | 0.228 | 0.127    | 0.144            | 0.102    |
|        | MAPE (%) | 180.9 | 144.9    | 192.9            | 125.9    |
| 6-hour | MAE      | 0.474 | 0.414    | 0.418            | 0.380    |
|        | MSE      | 0.474 | 0.364    | 0.347            | 0.270    |
|        | MAPE (%) | 310.2 | 200.7    | 249.2            | 187.3    |

Fig. 11 illustrates the error distributions derived from observed- predicted RST differentials, comparing model performance across 1-hour, 3-hour, and 6-hour forecasting intervals. At the 1-hour horizon, all models demonstrate relatively concentrated error distributions. The LSTM-stacking ensemble model achieves the narrowest error range spanning from -0.564 °C to 0.619 °C, marking a 29.3 % reduction relative to the baseline LSTM. Furthermore, it exhibits the smallest bias, with a mean error approaching 0 °C and a median of -0.002 °C, both approaching the theoretical optimum. In contrast, KNN-LSTM and Attention-BiLSTM demonstrate negative biases, indicating systematic overestimation of low-temperature risks.

At the 3-hour horizon, the LSTM-stacking ensemble model maintains the tightest error distribution spanning 3.021 °C, with its maximum positive error reaching 2.108 °C showing a 3.4 % reduction relative to Attention-BiLSTM, and its minimum negative error constrained at -0.913 °C demonstrating a 23.9 % improvement over KNN-LSTM. In the 6-hour forecast, the error range of the ensemble model is mitigated by 34.4 % compared to the LSTM. Its upper error bound reaching 3.189 °C is 37.1 % lower than that of LSTM, and it is the only model limiting negative errors within -1.4 °C, effectively suppressing the -2.492 °C prediction risk observed in Attention-BiLSTM under extreme cold conditions.

The LSTM model exhibits consistently inferior performance across all forecasting horizons, with a particularly concerning positive error spike of 5.066 °C observed in the 6-hour forecast, which may result in significant underestimation of RST values..

20




For short-term alerts, the LSTM-stacking ensemble model achieves near-zero bias control. In medium-term forecasts, prudent consideration is warranted due to the systematic overestimation of low-temperature conditions inherent in the KNN-LSTM variant. For long-term prediction, the error outliers of Attention-BiLSTM demand careful attention. The experimental results demonstrate that by strategically integrating the boundary control strength of KNN-LSTM with the mean stability of Attention-BiLSTM, the LSTM-stacking ensemble model provides an optimal solution for road temperature forecasting across multiple temporal scales.

Figure 11: Error distribution between predicted and observed winter RST values across 1-hour (a), 3-hour (b), 6-hour (c) forecasting intervals.

# 4.2 Winter RST prediction under sub-zero conditions

In the domain of winter traffic meteorology, special emphasis is placed on predictive performance under extreme sub-zero RST conditions. According to the expressway traffic meteorology, RST values below 0 °C are defined as critical low-temperature conditions (CMA, 2018), which may significantly compromise the operational efficiency and safety of expressway networks. A curated dataset of 1,338 extreme low-temperature samples meeting the RST 

|          | MSE      | 0.012 | 0.092 | 0.209 |
|----------|----------|-------|-------|-------|
|          | MAPE (%) | 41.9  | 191.0 | 247.3 |
|          | MAE      | 0.068 | 0.186 | 0.317 |
| Ensemble | MSE      | 0.008 | 0.058 | 0.153 |
|          | MAPE (%) | 35.0  | 94.5  | 186.7 |

Figure 12: Winter RST prediction under low-temperature conditions across 1-hour (a), 3-hour (b), and 6-hour (c) forecasting intervals.





As presented in Table 4, the LSTM-stacking ensemble model consistently outperforms individual variants across all error metrics under low-temperature conditions. For the 1-hour forecast horizon, the ensemble achieves the lowest MAE of 0.068, MSE of 0.008, and MAPE of 35.0%, demonstrating relative improvements of 24.4%, 42.9%, and 17.2%, respectively, compared to baseline LSTM. This performance advantage escalates at longer horizons: the 3-hour forecast interval reduces the mean absolute percentage error (MAPE) to 94.5%, markedly lower than the corresponding values of 162.6% obtained by the LSTM model and 191.0% obtained by the Attention-BiLSTM model. Even at 6-hour intervals with performance degradation, the ensemble maintains the lowest MSE value of 0.153 and MAPE value of 186.7%, preserving operational robustness amid increasing uncertainty.

The performance trends observed in Table 4 are further corroborated by Figure 12, where temporal evolution analysis demonstrates that the LSTM-stacking ensemble model precisely captures the amplitude and phasing of RST peaks and troughs, particularly during abrupt cold transitions. In contrast, the baseline LSTM exhibits significant deviations and temporal lag during low-temperature extremes, while KNN-LSTM and Attention-BiLSTM demonstrate pronounced tendencies toward oversmoothing or overreaction to short-term fluctuations. By synergizing the complementary strengths of constituent models, the ensemble framework achieves superior alignment with observational data and exhibits enhanced operational resilience under extreme low-temperature conditions.

# 435 4.3 Winter RST prediction under representative synoptic conditions

Historical meteorological and precipitation data were retrieved to identify two representative winter periods: stable synoptic conditions February 8-10, 2024 and overcast/rainy conditions February 23-25, 2024. The RST prediction performance of various models under these contrasting weather conditions was systematically analyzed. As shown in Fig. 13, under stable synoptic conditions, the presence of stable weather and strong solar radiation leads to more pronounced temperature fluctuations with clear periodic trends. Forecasting models demonstrate relatively high predictive accuracy during these stable periods, particularly over short time intervals, yielding smaller error margins. Conversely, during overcast and rainy conditions, increased relative humidity and precipitation induce more subtle, stochastic RST variations that lack clear periodicity. The models demonstrate constrained adaptability to such stochastic fluctuations, resulting in diminished predictive fidelity. This phenomenon becomes particularly pronounced during heavy precipitation events, where elevated temperature variability induces more substantial forecast deviations.

In summary, the disparity in model performance under clear versus rainy conditions underscores the meteorological sensitivity of winter RST dynamics. Specifically, under precipitation-dominated regimes, all models encounter significant challenges in maintaining both adaptability and predictive accuracy. The LSTM-stacking ensemble framework exhibits superior performance in these complex scenarios, underscoring its robust predictive capacity when confronted with non-stationary, and irregular thermal variations.

Figure 13: Winter RST prediction across and rainy and stable synoptic conditions across 1-hour (a, d), 3-hour (b, e), and 6-hour (c, f) forecasting intervals.

#### 455 **5 Conclusion**


This study focuses on winter RST time-series prediction through the development of an optimized LSTM model and a stacking-based ensemble strategy, thereby establishing a novel forecasting framework that has been empirically validated. Through systematic model development and optimization in this research, the following key conclusions emerge:

The KNN-LSTM variant enhances the capability of the model to recognize short-term climate patterns by incorporating similar historical samples. Meanwhile, the Attention-BiLSTM incorporates an attention-based temporal weighting mechanism, significantly improving responsiveness to critical temperature transitions. Compared to traditional LSTM, these two improved models demonstrate superior error mitigation and enhanced fitting performance, thereby confirming their efficacy in temporal modeling tasks. By integrating KNN-LSTM and Attention-BiLSTM models and using Bayesian ridge regression as a fusion technique, the improved LSTM-stacking ensemble model achieves superior error reduction and improved predictive alignment

- across multiple forecast horizons. In experiments replicating sub-zero RST regimes (





- Chen, Y. M. and Zhou, L. R.: Real-time bridge temperature prediction based on long short-term memory neural network and ground meteorological data, in: Proceedings of the 9th National Academic Conference on Structural Vibration Control and Health Monitoring, edited by: China Society of Vibration Engineering, South China University of Technology, 189, https://doi.org/10.26914/c.cnkihy.2023.113531, 2023.
  - China Meteorological Administration (CMA): Grade of Highway Traffic High-Impact Weather Warning, QX/T 414-2018, issued by China Meteorological Press, Beijing, 2018.
- 500 Cover, T. and Hart, P.: Nearest neighbor pattern classification, IEEE T. Inform. Theory, 13, 21-27, https://doi.org/10.1109/TIT. 1967.1053964, 1967.
  - Crevier, L. P. and Delage, Y.: METRo: A new model for road-condition forecasting in Canada, J. Appl. Meteorol., 40, 2026-2037, https://doi.org/10.1175/1520-0450(2001)040<2026:MANMFR>2.0.CO;2, 2001.
  - Dai, B., Yang, W., Ji, X., et al.: Winter highway hourly road surface temperature LSTM prediction model, Chin. J. Saf. Sci., 33, 136-145, https://doi.org/10.16265/j.cnki.issn1003-3033.2023.01.2215, 2023.
  - Gers, F. A., Schmidhuber, J., and Cummins, F.: Learning to forget: continual prediction with LSTM, Neural Computation, 12, 2451-2471, https://doi.org/10.1162/089976600300015015, 2000.
  - Guo, Y. H. and Wang, Z. D.: A multilevel optimal feature selection and ensemble learning for a specific CAD system-pulmonary nodule detection, Appl. Mech. Mater., 380-384, 1593-1599, https://doi.org/10.4028/www.scientific.net/AMM.380-384.1593, 2013.
  - Horne, J. E.: Thermal mapping and road-weather information systems for highway engineers, in: Highway Meteorology, edited by: Thornes, J. E., CRC Press, Boca Raton, FL, 39-68, 1991.
  - Hou, Y. Y., Zheng, E. R., Guo, W. Q., et al.: Temperature prediction of switchgear equipment based on long short-term memory recurrent neural network, Journal of Shaanxi University of Science and Technology, 39, 148-155, https://doi.org/10.19481/j.cnki.issn2096-398x.2021.04.023, 2021.
  - Ledezma, A., Aler, R., Sanchis, A., and Borrajo, D.: GA-stacking: evolutionary stacked generalization, Intell. Data Anal., 14, 89-119, https://doi.org/10.3233/IDA-2010-0410, 2010.
  - Li, Q. C., Xu, S. W., Zhang, Y. E., Hu, X. T., Wang, S. J., Zhang, J. H., Dong, G. Q., and Wang, K. R.: Stacking ensemble learning modeling and prediction of corn yield per unit area based on meteorological factors, Sci. Agr. Sinica, 57, 679-697, https://doi.org/10.3864/j.issn.0578-1752.2024.04.005, 2024.
  - Li, W.: A Hybrid Method for Winter Road Surface Temperature Prediction Using Improved LSTMs and Stacking-Based Ensemble Learning, Zenodo, https://doi.org/10.5281/zenodo.17312485, 2025.
  - Luo, X., Li, D., Yang, Y., Zhang, S., Cao, L., and Zhang, J.: Spatiotemporal traffic flow prediction with KNN and LSTM, J. Adv. Transport., 2019, 4145353, https://doi.org/10.1155/2019/4145353, 2019.
- Nowrin, T. and Kwon, T. J.: Forecasting short-term road surface temperatures considering forecasting horizon and geographical attributes an ANN-based approach, Cold Reg. Sci. Technol., 202, 103631, https://doi.org/10.1016/j.coldregions.2022.10 3631, 2022.

- Schuster, M., and Paliwal, K. K.: Bidirectional recurrent neural networks, IEEE trans. Signal Process., 45(11), 2673-2681, 451 https://doi.org/10.1109/78.650093, 1997.
- Shao, J. and Lister, P. J.: An automated nowcasting model of road surface temperature and state for winter road maintenance, J. Appl. Meteorol., 35, 1352-1361, https://www.jstor.org/stable/26188273, 1996.
  - Spearman, C.: The proof and measurement of association between two things, in: Studies in Individual Differences: The Search for Intelligence, edited by: Jenkins, J. J. and Paterson, D. G., Appleton Century Crofts, 45-58, https://doi.org/10.1037/114 91-005, 1961.
- Subramanian, D., Subramanian, V., Deswal, A., Mann, D. L., and Bozkurt, B.: New predictive models of heart failure mortality using time-series measurements and ensemble models, Circ.-Heart Fail., 4, 456-462, https://doi.org/10.1161/CIRCHEART FAILURE.110.958496, 2011.
  - Wolpert, D. H.: Stacked generalization, Neural Networks, 5, 241-259, https://doi.org/10.1016/S0893-6080(05)80023-1, 1992.
- Xie, R., Liao, C., Luo, X., et al.: Research on road surface temperature characteristics and road ice warning model of ordinary highways in winter in Hunan Province, central China, Front. Earth Sci., 11, 1251635, https://doi.org/10.3389/feart.2023.12 51635, 2023.
  - Yan, X., Tie, C. C., Yan, W., et al.: Global temperature prediction analysis based on ARIMA model and CNN-LSTM combined model, Science and Technology & Innovation, 2, 19-22, https://doi.org/10.15913/j.cnki.kjycx.2024.02.005, 2024.
- Yang, G. Q., Liu, S. L., Wang, D. Y., Hao, Z. J., and Wang, L.: Short-term wind power prediction based on Attention-GRU wind speed correction and Stacking, Acta Energ. Sol. Sin., 43, 273-281, https://doi.org/10.19912/j.0254-0096.tynxb.2021-0712, 2022.
  - Yang, Y. and Chen, K.: Temporal data clustering via weighted clustering ensemble with different representations, IEEE T. Knowl. Data En., 23, 307-320, https://doi.org/10.1109/TKDE.2010.112, 2010.
- Zhang, M., Guo, H., Li, J., et al.: A deep learning approach for enhanced real-time prediction of winter road surface temperatures in high-altitude mountain areas, Promet-Traffic&Transportation, 36, 958-972, https://doi.org/10.7307/ptt.v36i5.541, 2024.
  - Zheng, C., Kong, Y. H., Kong, X. Y., Zhou, T. H., and Cao, Y. H.: Power load forecasting based on ensemble learning GBDT algorithm, in: Proceedings of the 2024 Academic Annual Conference of Jilin Electric Power Engineering Society, edited by: Jilin Electric Power Engineering Society, State Grid Yanbian Power Supply Company, 279-285, https://doi.org/10.26914/c.cnkihy.2024.033674, 2024.
- Zhou, B., Wen, Q. H., Xu, Y., Song, L., and Zhang, X.: Projected changes in temperature and precipitation extremes in China by the CMIP5 multimodel ensembles, J. Climate, 27, 6591-6611, https://doi.org/10.1175/JCLI-D-13-00761.1, 2014.
  - Zhou, W. Y., Liu, L. L., and Zhang, Z. Y.: Semantic relation extraction combining multi-layer attention mechanism and bidirectional LSTM, Software Guide, 18, 10-14, https://doi.org/10.11907/rjdk.182763, 2019.