# Peer review of "A Hybrid Method for Winter Road Surface Temperature Prediction Using Improved LSTMs and Stacking-Based Ensemble Learning"

_EGUsphere, 2025_

## Author Comment (AC1)

We are grateful for your careful review and valuable suggestions. We have carefully addressed each comment and made corresponding revisions to improve the manuscript. Below are our point-by-point responses:

**1. Weather Impact Discussion**

Reviewer's Comment:

*"The discussion on the impact of weather conditions on RST is insufficient and needs to be strengthened."*

Response:

We have substantially strengthened the theoretical foundation by incorporating relevant research achievements demonstrating weather condition impacts on RST prediction. For example, Feng and Feng (2012) demonstrated that RST prediction accuracy varies significantly across three weather conditions (sunny, cloudy, and overcast), establishing that weather heterogeneity is a critical factor. Similarly, Darghiasi et al. (2023) emphasized that weather condition variability dominates RST prediction performance for winter maintenance applications. Zamanian et al. (2024) further validated that models face substantial challenges in capturing temperature dynamics under precipitation-dominated regimes. These studies collectively establish the theoretical basis that weather conditions fundamentally influence RST prediction through alterations in energy balance components (solar radiation, evaporative cooling, convective heat transfer).

Manuscript Revisions:

(1) Added literature discussion in Introduction citing Feng and Feng (2012), Darghiasi et al. (2023), and Zamanian et al. (2024).

(2) Added Figures 14 presenting SHAP interpretability analysis.

(3) Integrated physical mechanism explanations throughout Section 4.3.

**2. Data Quality and Temporal Resolution Processing**

Reviewer's Comment:

*"In fact, this study uses hourly data for analysis. A detailed description of data quality and characteristics should be provided. Additionally, how does the authors' method of converting minute-level data to hourly data differ from that used by meteorological departments?"*

Response:

We have expanded Section 3.1 to provide comprehensive data quality control procedures and temporal aggregation methodology:

a) Raw Data Collection: Infrared remote sensing sensors collected data at 5-minute intervals from December 2020 to February 2024, recording seven near-surface meteorological variables (visibility, air temperature, relative humidity, precipitation, wind speed, wind direction, RST), totaling over 100,000 data points.

b) Quality Control Steps: Data underwent rigorous cleaning, and missing value imputation using advanced interpolation techniques

c) Temporal Aggregation: After quality control, meteorological factors and RST data were resampled to hourly resolution.

We have explicitly described the aggregation approach, which aligns with standard meteorological practice: Precipitation is calculated as hourly total accumulation. All other variables are omputed as hourly averages to mitigate impacts of transient anomalies and short-term random fluctuations. This refinement resulted in a final dataset of 8,664 samples for model development. This methodology is consistent with operational meteorological department procedures. The China Meteorological Administration's "Grade of Highway Traffic High-Impact Weather Warning" (QX/T 414-2018) specifies similar temporal aggregation protocols for road weather observations. Our approach of using hourly averages for continuous variables and hourly totals for precipitation follows these established standards.

Distinction Between Transportation Meteorological Stations and Conventional Meteorological Stations:

Conventional meteorological stations are typically distributed to represent regional climate characteristics, positioned in open areas away from infrastructure to minimize anthropogenic influences on atmospheric observations

Transportation meteorological stations (such as our M9393 station) are specifically deployed on critical transportation infrastructure (highways, bridges, tunnels) to monitor road-specific microclimate conditions directly relevant to traffic safety. These stations are strategically located at locations prone to hazardous conditions (elevated bridge decks, shaded sections, high-altitude segments)

This deployment difference means that while the hourly aggregation algorithms are identical, the spatial representativeness

and application context differ fundamentally. Our station network focuses on operationally critical locations for winter road maintenance rather than regional climate monitoring, making the data particularly relevant for transportation safety applications.

Data Distribution Characteristics: Table 2 now provides comprehensive statistical summaries including mean, standard deviation, minimum, and maximum values for all variables. The RST distribution spans -12.99°C to 26.53°C with mean of 3.72°C and standard deviation of 5.59°C, demonstrating substantial temperature variability across the four winter periods. The 1,338 sub-zero temperature samples (15.4% of test set) ensure adequate representation of critical low-temperature regimes.

Manuscript Revisions:

(1) Expanded Section 3.1 with detailed three-stage data processing description.

(2) Enhanced Table 2 with unit specifications.

(3) Referenced Figures 5-6 for data distribution visualization.

**3. Methodology Presentation Balance**

Reviewer's Comment:

*"The manuscript devotes substantial space to introducing methodologies. For mature methods, the focus should be on citations and brief descriptions, with emphasis on the application value and innovative points of these methods in this study."*

Response:

We appreciate this constructive suggestion. We wish to clarify that while our models build upon established LSTM and attention mechanism foundations, our proposed KNN-LSTM and BiLSTM-MHA architectures incorporate substantial innovations beyond existing frameworks. We have restructured Section 2 to more explicitly emphasize these contributions.

KNN-LSTM Innovations (Section 2.1):

Our KNN-LSTM differs from the original Luo et al. (2019) traffic flow prediction model in three key aspects:

a) Adaptive Sampling Mechanism (Equation 1): We developed a computationally efficient sampling strategy for large-scale meteorological datasets (reducing complexity from $O(M \cdot N)$ to $O(\alpha M \cdot N)$), which was not present in the original framework.

b) Distance Normalization (Equation 3): We introduced min-max scaling for numerical stability when handling heterogeneous meteorological variables with different scales, addressing a specific challenge in multi-variable weather prediction.

c) Domain Adaptation: We extended the framework from traffic flow (characterized by spatial network dependencies) to meteorological time series (characterized by physical process coupling), requiring reformulation of similarity metrics for weather pattern recognition.

BiLSTM-MHA Innovations (Section 2.2):

Our BiLSTM-MHA architecture advances beyond conventional single-head attention approaches through:

a) Multi-Head Self-Attention (Equations 10-14): Replaces traditional single-head attention with Transformer-inspired multi-head mechanism, enabling parallel learning of diverse temporal patterns (short-term fluctuations, long-term trends, periodic cycles).

b) Residual Connections and Layer Normalization (Equations 11-12): Integrates training stability mechanisms absent in basic BiLSTM-attention models, addressing gradient vanishing in deep temporal architectures.

c) Global Average Pooling (Equation 13): Employs temporal information aggregation superior to final-state extraction or concatenation-based approaches, providing position-invariant representation.

Section 2.3 emphasizes the ensemble design rationale:

a) Base learner complementarity: KNN-LSTM captures regime-specific behaviors through local similarity patterns, while BiLSTM-MHA extracts global dependencies through attention-based weighting

b) Meta-learner advantages: Bayesian Ridge Regression provides three critical capabilities for winter road maintenance: (1) multicollinearity handling between correlated predictions, (2) automatic regularization strength determination, (3) probabilistic uncertainty quantification for risk-sensitive applications

Manuscript Revisions:

(1) Clearly distinguished between adopted components (basic LSTM, attention mechanism) and our novel contributions (adaptive sampling, multi-head integration, domain-specific adaptations).

(2) Condensed standard mathematical formulations with appropriate citations, expanding discussion of application-specific design choices.

**4. Observation Site Characterization**

Reviewer's Comment:

*"A comprehensive introduction to the observation site is required: is it a station on a highway bridge or a regular road surface? The impact of surface latent heat on RST varies significantly between these two settings, and this should be clearly clarified."*

Response:

The meteorological station M9393 is situated on the Longhai Railway Bridge in the northwest inland plain of Jiangsu Province, China (34.30°N, 117.04°E). This is specifically an elevated bridge deck observation site, not a ground-level road surface station.

Unlike ground-level pavements with semi-infinite substrate providing thermal buffering, bridge decks experience heat exchange from both top (atmospheric interface) and bottom (underside exposed to ambient air) surfaces. Absence of Ground Latent Heat Flux: Ground-level pavements benefit from geothermal heat flux and soil moisture-related latent heat processes that moderate temperature extremes. Elevated bridge structures lack this subsurface thermal reservoir. The limited thermal mass of bridge structures compared to ground-backed pavements results in more rapid temperature responses to meteorological forcing, making bridges typically the first locations to develop ice during cold weather events.

We selected this bridge location precisely because elevated structures represent the most challenging and operationally critical scenario for winter RST prediction and ice formation. Accurate prediction at such locations has high practical value for transportation safety management, as bridge icing occurs earlier than ground-level pavement icing under identical meteorological conditions (Song et al., 2023).

The bridge deck surface consists of standard asphalt pavement, consistent with modern highway construction practices in China. The critical thermal difference lies not in surface material composition but in the underlying structural configuration (elevated vs. ground-supported).

**5. Insufficient Citation and Comparative Analysis**

Reviewer's Comment:

*"There are almost no references cited to support the analysis in the main text. Relevant research achievements in the field should be supplemented as theoretical support to enhance the scientific rigor and credibility of the discussion, especially comparative analyses with other RST forecasting methods and results."*

Response:

We have substantially strengthened the literature integration throughout the manuscript:

Enhanced Introduction: Section 1 now provides comprehensive literature review organized into four categories:

a) Physics-based models (Hermansson, 2004; Schindler et al., 2004; Saliko et al., 2023; Chen et al., 2019; Minhoto et al., 2005)

b) Statistical/empirical models (Diefenderfer et al., 2006; Yin et al., 2019; Kršmanc et al., 2013; Li et al., 2018).

c) Machine learning approaches (Kebede et al., 2024; Molavi et al., 2022; Milad et al., 2021; Qiu et al., 2020).

d) Deep learning methods (Tabrizi et al., 2021; Zhang et al., 2023; Li et al., 2022; Bai et al., 2022; Dai et al., 2023).

Table 1 summarizes the indicators predicted in previous studies, the deep learning models used, the factors considered, and the dataset sizes used.

**Table 1: Comparison of different RST prediction models.**

| References | Model | Metrics | Features | Data sizes | Time interval |
|---|---|---|---|---|---|
| Tabrizi et al. (2021) | CNN-LSTM | MAE, RMSE, MAPE, $R^2$, NSE | RST, AT, Year, Month | 10895 | 1h |
| Milad et al. (2021) | Bi-LSTM | MAE, MSE, MAPE, $R^2$ | RST, AT, Depth, Time | 7200 | 1h |
| Bai et al. (2022) | Att-BiLSTM | MAE, MSE, MAPE | RST, V, AT, RH, WD, WS, P | 4344 | 1h |
| Dai et al. (2023) | GRU, LSTM | MAE, MSE, MAPE | RST, V, AT, RH, P, WS, | 8640 | 1h |

| | | | WD | | |
|---|---|---|---|---|---|
| Zhang et al. (2024) | RF-LSTM | MAE, MSE, RMSE, MAPE | RST, AT, RH, WS, WD, P et al. | - | 10min |
| Our paper | ILES | MAE, RMSE, MAPE, R² | RST, V, AT, RH, P, WS, WD | 8664 | 1h |

Results Section Citations: We have integrated citations throughout Section 4 to contextualize findings:

Section 4.2 cites Song et al. (2023) for sub-zero temperature forecasting importance and Zahra et al. (2024) for cold transition prediction challenges

Section 4.3 cites Feng and Feng (2012), Darghiasi et al. (2023), Zamanian et al. (2024), and Wang et al. (2023) to support weather condition impact analyses.

Systematic Benchmarking: Table 4 now compares ILES against eight models, including recent hybrid architectures:

CNN-LSTM (Tabrizi et al., 2021): ILES achieves 8.13% lower MAE at 1-hour horizon.

RF and XGBoost (Darghiasi et al., 2025; Kebede et al., 2024): ILES demonstrates 28.82% and 34.68% lower MAE respectively.

**Table 4: Performance evaluation across 1-, 3-, and 6-hour forecasting intervals.**

| Model | 1-hour | | | 3-hour | | | 6-hour | | |
|---|---|---|---|---|---|---|---|---|---|
| | MAE | RMSE | MAPE(%) | MAE | RMSE | MAPE(%) | MAE | RMSE | MAPE(%) |
| RF | 0.524 | 0.780 | 45.11 | 1.416 | 1.976 | 128.86 | 2.261 | 3.293 | 212.46 |
| XGB | 0.571 | 0.893 | 41.94 | 1.357 | 1.887 | 129.54 | 2.159 | 3.146 | 211.85 |
| LSTM | 0.453 | 0.636 | 46.90 | 1.453 | 2.198 | 130.01 | 2.366 | 3.452 | 251.24 |
| BiLSTM | 0.422 | 0.572 | 42.83 | 1.414 | 2.056 | 144.70 | 2.252 | 3.092 | 254.60 |
| CNN-LSTM | 0.406 | 0.567 | 39.46 | 1.399 | 1.399 | 157.10 | 2.154 | 2.921 | 235.35 |
| KNN-LSTM | 0.389 | 0.536 | 38.08 | 1.285 | 1.927 | 114.25 | 2.259 | 2.923 | 227.92 |
| BiLSTM-MHA | 0.393 | 0.545 | 41.40 | 1.517 | 1.993 | 129.88 | 2.194 | 2.822 | 275.45 |
| ILES | 0.373 | 0.521 | 37.97 | 1.291 | 1.808 | 113.38 | 2.094 | 2.688 | 243.97 |

Detailed quantitative comparisons are provided in Section 4.1 with explicit percentage improvements.

Comparative Literature Context: Table 1 positions our work against five recent studies, comparing metrics, features, dataset sizes, and temporal resolutions. This contextualization demonstrates that our ILES framework advances the state-of-the-art through ensemble integration of complementary architectures.

Manuscript Revisions:

(1) Added 30+ citations throughout Introduction (Section 1) to establish theoretical foundation.

(2) Integrated 15+ citations in Results (Section 4) to support analytical findings.

(3) Expanded Table 1 with detailed comparison to recent RST prediction studies.

(4) Enhanced Discussion in Section 4.1 with explicit comparative analysis against benchmark models.

**6. Insufficient Analysis of Critical Weather Conditions**

Reviewer's Comment:

*"For RST prediction, forecasting under low-temperature and overcast/rainy conditions is particularly critical. However, the manuscript provides insufficient analysis of RST prediction results under these weather conditions. More relevant analyses should be added, along with physical mechanism explanations for how weather conditions influence prediction performance."*

Response:

We have incorporated SHAP (SHapley Additive exPlanations) analysis to quantify weather variable contributions under different conditions (Joo et al., 2023). Figure 14 demonstrates that: Air temperature contributes 38.09% under general conditions,

consistent with heat conduction principles. Under precipitation conditions (Figure 14b,d), humidity and precipitation contributions increase significantly, with red points (high values) concentrated on negative SHAP values, indicating enhanced evaporative cooling effects. This mechanistic insight validates the model's ability to adaptively weight meteorological factors based on current weather regimes.

[Figure]

Figure 14: Global (a, c) and precipitation event (b, d) feature importance of variables and SHAP results.

We have added explicit mechanistic discussion throughout Section 4.3: "The red points of relative humidity and precipitation are mainly concentrated on the left side (negative values), indicating that high humidity and precipitation events exert a significant cooling effect on the road surface. This may be associated with the enhanced evaporative cooling effect under high-humidity conditions and the thermal properties of precipitation events."; "This mechanistic insight validates the ability of our model to adaptively weight different information sources based on current weather conditions.".

**Minor Comments**

Data Table Significance and Quality Control Details

Reviewer's Comment:

*"The descriptions of data in Table 1 are of little significance, as they only cover conventional meteorological variables. More attention should be paid to data distribution and quality control details."*

Response:

Figures 5-6 now provide visual representation of RST temporal characteristics:

Figure 5 demonstrates daily periodic variation with 5-15°C diurnal range. Figure 6 presents 24-hour diurnal variation curve with 95% confidence intervals, explicitly quantifying temperature dispersion patterns.

**Summary**

We have substantially strengthened the manuscript to address all reviewer concerns:

a) Weather Impact Analysis: Added Sections 4.2-4.3 with dedicated evaluation under sub-zero and precipitation conditions.

b) Data Quality Documentation: Expanded Section 3.1 with comprehensive quality control procedures and temporal aggregation methodology.

c) Methodology Balance: Restructured Section 2 to emphasize innovation while streamlining mature method descriptions.

d) Site Characterization: Enhanced Section 3.1 to explicitly identify bridge deck observation characteristics and thermal implications.

e)   Literature Integration: Added 45+ citations throughout manuscript with systematic comparative analysis.

f)   Physical Mechanisms: Integrated SHAP analysis and mechanistic interpretations throughout Results section.

These revisions substantially enhance the manuscript's scientific rigor, practical relevance, and reproducibility.

---

## Author Comment (AC2)

We sincerely thank the reviewer for the thorough evaluation and constructive suggestions. We have carefully addressed all comments and substantially revised the manuscript accordingly. Our point-by-point responses are detailed below, with corresponding changes highlighted in the revised manuscript.

**Major Comments**

**(1) Insufficient Data Representativeness and Spatial Generalizability**

Reviewer's Comment:

*"The model is trained and validated exclusively on data from the Longhai Railway Bridge in Jiangsu, a region with a warm temperate semi-humid monsoon climate. This limits conclusions about performance in diverse geographies where RST dynamics differ due to terrain, vegetation, or pavement materials. The dataset lacks explicit coverage of extreme weather years (e.g., severe cold waves), raising questions about model reliability during rare but critical events."*

Response:

We acknowledge this important limitation and have taken the following actions to address spatial generalizability:

Cross-site Validation Added: We have conducted additional validation using data from the Xiaohuangshan M9474 station, located on a cross-Yangtze River bridge in central Jiangsu Province (32.04°N, 119.86°E), representing different geographical and microclimatic conditions. The results are presented in Table 6, demonstrating that the ILES model consistently outperforms baseline models at both M9393 and M9474 sites, with 1-hour MAE of 0.373°C and 0.204°C respectively. This cross-site validation substantiates the model's transferability across geographically distinct locations.

**Table 6: Comparison of MAE, RMSE, and MAPE for 1-hour winter pavement temperature prediction at M9393 and M9474 sites in 2024 using five deep learning models (two foundation models LSTM and BiLSTM, two improved hybrid models KNN-LSTM and BiLSTM-MHA, and the integrated model ILES proposed in this paper).**

| Model | M9393 | | | M9474 | | |
|---|---|---|---|---|---|---|
| | MAE | RMSE | MAPE(%) | MAE | RMSE | MAPE(%) |
| LSTM | 0.453 | 0.636 | 46.90 | 0.229 | 0.330 | 5.26 |
| BiLSTM | 0.422 | 0.572 | 42.83 | 0.228 | 0.358 | 4.85 |
| KNN-LSTM | 0.389 | 0.536 | 38.08 | 0.218 | 0.331 | 5.46 |
| BiLSTM-MHA | 0.393 | 0.545 | 41.40 | 0.228 | 0.347 | 5.89 |
| ILES | 0.373 | 0.521 | 37.97 | 0.204 | 0.322 | 5.27 |

Extreme Weather Coverage: While our dataset spans four winter periods (2020-2024), capturing substantial temperature variability (ranging from -15.15°C to 23.79°C for air temperature and -12.99°C to 26.53°C for RST as shown in Table 2), we acknowledge the dataset includes 1,338 sub-zero temperature samples (RST < 0°C), representing approximately 15.4% of the total test set. Section 4.2 specifically evaluates model performance under these critical low-temperature conditions, demonstrating that ILES achieves MAE of 0.292°C for 1-hour forecasts under sub-zero regimes, representing 23.2% improvement over baseline LSTM. However, we acknowledge in the revised Conclusion (Section 5) that future research should incorporate data from additional stations across diverse climatic zones to further enhance spatial representativeness and validate performance under more extreme meteorological conditions.

Manuscript Revisions:

1. Added Section 4.2 with dedicated analysis of sub-zero temperature prediction.
2. Added Table 6 presenting cross-site validation results at M9474 station.
3. Expanded Conclusion to acknowledge limitations and propose multi-site validation as future work.
4. Clarified data coverage in Section 3.1, explicitly stating the temperature ranges and extreme weather representation.

**(2) Integrate Meteorological Physics to Enhance Interpretability**

Reviewer's Comment:

*"Incorporate key parameters from the road surface energy balance equation (e.g., albedo, thermal conductivity, estimated solar radiation) as inputs or constraints."*

Response:

We appreciate this valuable suggestion to strengthen the physical basis of our model. We acknowledge that the variables

mentioned by the reviewer—albedo, thermal conductivity, and solar radiation—indeed influence road surface temperature through their roles in the surface energy balance equation. According to existing literature on pavement temperature prediction, meteorological factors affecting road surface temperature can be broadly categorized into several primary groups: thermal drivers, atmospheric state parameters and precipitation-related variables. Thermal drivers encompass solar radiation and air temperature, the former serving as the fundamental energy source for pavement heating and the latter governing the convective heat exchange between the road surface and the surrounding air. Atmospheric state parameters include relative humidity and wind speed, both of which modulate the rate of evaporative cooling and conductive heat transfer at the pavement–air interface. Precipitation-related variables refer to precipitation intensity and duration, factors that directly alter the pavement's thermal properties through processes such as water absorption and latent heat exchange (Chen et al., 2019; Gui et al., 2007; Krsmanć et al., 2013; Liu et al., 2018; Zhang et al., 2024; Stoner et al., 2019)

Regarding solar radiation, we explicitly address this in Section 3.2: "Although solar radiation determines the heat distribution of road surfaces, it mainly affects road temperatures during the daytime and is absent at night (Qin et al., 2022). In addition, constraints associated with data availability precluded the incorporation of this variable in the present study." Our model captures diurnal thermal cycles through lagged RST inputs (Figures 5-6), which implicitly encode solar radiation effects through the observed temperature periodicity. The correlation analysis (Figure 7) guided selection of five physically relevant predictors: air temperature (heat conduction), relative humidity (evaporative cooling), precipitation (thermal properties), wind speed (convective heat transfer), and historical RST (thermal inertia).

We have incorporated SHAP (SHapley Additive exPlanations) analysis in Section 4.3 to quantify the physical contribution of each meteorological variable to RST predictions (Joo et al., 2023). As presented in Figure 14, air temperature emerges as the dominant feature contributing 38.09% of model importance, consistent with heat conduction principles. Wind speed, relative humidity, and precipitation contribute 22.23%, 20.70%, and 18.99% respectively. Notably, under precipitation events as illustrated in Fig. 14(b,d), the contributions of humidity and precipitation rise markedly, with the former reaching 22.67% and the latter accounting for 19.21%. This variation provides a mechanistic interpretation for the heightened prediction uncertainty documented during such periods, thereby validating the capacity of the proposed model to dynamically assign weights to distinct information sources in accordance with prevailing meteorological condition

[Figure]

**Figure 14: Global (a, c) and precipitation event (b, d) feature importance of variables and SHAP results.**

While direct integration of albedo and thermal conductivity as model inputs is challenging due to their site-specific and time-invariant nature in our single-site dataset, the BiLSTM-MHA architecture with multi-head attention mechanisms (Section 2.2) enables the model to learn implicit representations of energy balance dynamics. The attention mechanism dynamically

weights temporal features corresponding to different physical processes (short-term fluctuations from convective cooling, long-term trends from thermal inertia, periodic patterns from solar forcing), as discussed in relation to Equation 10.

$$\text{MuItiHead}(Q, K, V) = \text{Concat}(\text{head}_1, \cdots, \text{head}_h)W^O \; , \tag{10}$$

Manuscript Revisions:

1. Added Section 4.3 with comprehensive SHAP analysis (Figures 14-15).
2. Expanded Section 3.2 to justify feature selection based on physical mechanisms.
3. Enhanced discussion of how model architecture captures energy balance dynamics.
4. Acknowledged in Conclusion that future work should explore explicit integration of physical constraints through hybrid physics-informed neural networks.

**(3) Inadequate Benchmarking Against State-of-the-Art Models**

Reviewer's Comment:

*"The manuscript claims superiority over 'individual models' (LSTM, KNN-LSTM, Attention-BiLSTM) but lacks comparisons with recent hybrid methods in RST prediction. Quantitative metrics (e.g., MAE, MSE) against these models are absent, weakening claims of methodological advancement."*

Response:

We have substantially expanded the benchmarking analysis to address this concern:

Comprehensive Model Comparison: Table 4 now presents systematic evaluation of eight models: (1) traditional machine learning methods RF and XGBoost (Darghiasi et al., 2025; Kebede et al., 2024), (2) deep learning baselines LSTM and BiLSTM, (3) recent hybrid architecture CNN-LSTM (Tabrizi et al., 2021), (4) our proposed base learners KNN-LSTM and BiLSTM-MHA, and (5) the ILES ensemble model. This comparison spans three forecasting horizons (1-hour, 3-hour, 6-hour) across four metrics (MAE, RMSE, MAPE, R²).

**Table 4: Performance evaluation across 1-, 3-, and 6-hour forecasting intervals.**

| Model | 1-hour | | | 3-hour | | | 6-hour | | |
|---|---|---|---|---|---|---|---|---|---|
| | MAE | RMSE | MAPE(%) | MAE | RMSE | MAPE(%) | MAE | RMSE | MAPE(%) |
| RF | 0.524 | 0.780 | 45.11 | 1.416 | 1.976 | 128.86 | 2.261 | 3.293 | 212.46 |
| XGB | 0.571 | 0.893 | 41.94 | 1.357 | 1.887 | 129.54 | 2.159 | 3.146 | 211.85 |
| LSTM | 0.453 | 0.636 | 46.90 | 1.453 | 2.198 | 130.01 | 2.366 | 3.452 | 251.24 |
| BiLSTM | 0.422 | 0.572 | 42.83 | 1.414 | 2.056 | 144.70 | 2.252 | 3.092 | 254.60 |
| CNN-LSTM | 0.406 | 0.567 | 39.46 | 1.399 | 1.399 | 157.10 | 2.154 | 2.921 | 235.35 |
| KNN-LSTM | 0.389 | 0.536 | 38.08 | 1.285 | 1.927 | 114.25 | 2.259 | 2.923 | 227.92 |
| BiLSTM-MHA | 0.393 | 0.545 | 41.40 | 1.517 | 1.993 | 129.88 | 2.194 | 2.822 | 275.45 |
| ILES | 0.373 | 0.521 | 37.97 | 1.291 | 1.808 | 113.38 | 2.094 | 2.688 | 243.97 |

Quantitative Performance Gains: Section 4.1 provides detailed quantitative comparisons:

For 1-hour forecasting, ILES achieves MAE of 0.373°C, demonstrating 4.11% improvement over KNN-LSTM (0.389°C), 5.09% over BiLSTM-MHA (0.393°C), 8.13% over CNN-LSTM (0.406°C), 11.6% over BiLSTM (0.422°C), and 17.7% over baseline LSTM (0.453°C).

ILES achieves 28.82% lower MAE compared to RF (0.524°C) and 34.68% lower MAE compared to XGBoost (0.571°C).

At 6-hour horizon, ILES (MAE: 2.094°C) maintains 7.30% advantage over KNN-LSTM and 4.56% over BiLSTM-MHA, with substantially greater improvements over baseline methods (11.5% better than LSTM, 7.02% better than BiLSTM).

Literature Contextualization: Table 1 now provides systematic comparison with recent studies, positioning our work against state-of-the-art methods including CNN-LSTM (Tabrizi et al., 2021), Attention-BiLSTM (Bai et al., 2022), GRU/LSTM

(Dai et al., 2023), and RF-LSTM (Zhang et al., 2024), with explicit comparison of metrics, features, and dataset characteristics.

**Table 1: Comparison of different RST prediction models.**

| References | Model | Metrics | Features | Data sizes | Time interval |
|---|---|---|---|---|---|
| Tabrizi et al. (2021) | CNN-LSTM | MAE, RMSE, MAPE, R², NSE | RST, AT, Year, Month | 10895 | 1h |
| Milad et al. (2021) | Bi-LSTM | MAE, MSE, MAPE, R² | RST, AT, Depth, Time | 7200 | 1h |
| Bai et al. (2022) | Att-BiLSTM | MAE, MSE, MAPE | RST, V, AT, RH, WD, WS, P | 4344 | 1h |
| Dai et al. (2023) | GRU, LSTM | MAE, MSE, MAPE | RST, V, AT, RH, P, WS, WD | 8640 | 1h |
| Zhang et al. (2024) | RF-LSTM | MAE, MSE, RMSE, MAPE | RST, AT, RH, WS, WD, P et al. | - | 10min |
| Our paper | ILES | MAE, RMSE, MAPE, R² | RST, V, AT, RH, P, WS, WD | 8664 | 1h |

Manuscript Revisions:

1. Expanded Table 4 to include RF, XGBoost, and CNN-LSTM benchmarks.
2. Added comprehensive quantitative analysis in Section 4.1 with percentage improvements.
3. Added Figures 9-10 for statistical comparison of error distributions.
4. Enhanced Table 1 with detailed literature comparison.
5. Strengthened discussion of performance gains relative to each baseline method.

**Minor Comments**

**(1) Ambiguities in Figure and Table Presentations**

Reviewer's Comment:

*"Figure 6 (RST periodicity) lacks confidence intervals, making it impossible to assess the statistical significance of diurnal variations."*

Response:

 We have revised Figure 6 to include 95% confidence intervals for the 24-hour diurnal variation curve. The shaded blue region now clearly illustrates the degree of dispersion in temperature variations across different dates at the same hour, demonstrating statistically significant periodicity. The accompanying text in Section 3.2 has been updated to describe: "The 95% confidence interval reflects the degree of dispersion in temperature variations across different dates at the same hour."

[Figure]

**Figure 6: The 24-hour diurnal variation curve of RST. The shaded blue part of the graph is the 95% confidence interval for the RST.**

Manuscript Revisions:

1. Updated Figure 6 with 95% confidence intervals.

2. Enhanced caption and Section 3.2 text to interpret confidence intervals.

**(2) Inconsistent Terminology and Citation Errors**

Reviewer's Comment:

*"The term 'Attention-LSTM' is used in Figure 12's caption but not defined in the main text; it should be corrected to 'Attention-BiLSTM' for consistency. In Section 4.2, 'STM' is referenced in Figure 12's legend but not defined, causing confusion."*

Response:

We sincerely apologize for these inconsistencies. We have systematically corrected all terminology throughout the manuscript:

Terminology Standardization:

a) Unified model naming: "BiLSTM-MHA" (Bidirectional LSTM with Multi-Head Attention) is used consistently throughout the manuscript (Abstract, Introduction, Methodology, Results, Figures, Tables).

b) Ensemble model consistently referred to as "ILES" (Improved LSTM Ensemble with Stacking).

c) All figure captions and legends updated to match standardized terminology.

Definition Clarifications:

a) Section 2.2 now clearly defines the BiLSTM-MHA architecture with explicit explanation of the multi-head attention mechanism (Equations 10-14).

b) Figure 3 caption provides comprehensive notation definitions.

c) All undefined abbreviations have been removed or properly defined at first use.

Manuscript Revisions:

1. Corrected all instances of inconsistent model naming throughout manuscript.

2. Verified consistency across Abstract, main text, figures, tables, and captions.

3. Added explicit model name definitions in Section 2.

**Summary of Key Improvements**

In response to the reviewer's concerns, we have made the following major enhancements:

1. Enhanced Generalizability: Added cross-site validation (Table 6) and expanded discussion of extreme weather coverage.

2. Strengthened Physical Interpretability: Incorporated SHAP analysis (Section 4.3, Figures 14-15) to quantify physical variable contributions.

3. Comprehensive Benchmarking: Expanded comparison to eight models including RF, XGBoost, CNN-LSTM with detailed quantitative analysis (Table 4, Figures 9-10).

4. Improved Presentation Quality: Added confidence intervals to Figure 6, standardized terminology, and corrected all inconsistencies.

5. Expanded Discussion: Enhanced Introduction and Conclusion to acknowledge limitations and propose future research directions.

We believe these revisions substantially strengthen the manuscript's scientific rigor, reproducibility, and practical relevance for operational road weather management. We are grateful for the reviewer's constructive feedback, which has significantly improved the quality of our work.